# Construction of a Medical Radiation-Shielding Environment by Analyzing the Weaving Characteristics and Shielding Performance of Shielding Fibers Using X-ray-Impermeable Materials

Seon-Chil Kim

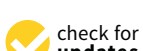



Department of Biomedical Engineering, School of Medicine, Keimyung University, 1095 Dalgubeol-daero, Daegu 42601, Korea; chil@kmu.ac.kr; Tel.: +82-10-4803-7773

**Abstract:** As the scope of radiation use in medical and industrial fields has expanded, interest in radiation shielding is increasing. Most existing radiation shields use Pb-based products, primarily in the form of a laminated sheet, which requires attention as fine cracks may occur depending on the usage and storage conditions. The weight of the sheets limits users' activities, and they pose a risk of heavy metal contamination. To address these problems, this study proposed a shielding fiber with improved flexibility and workability, and thus, produce a shielding garment. Masterbatches of polyethylene terephthalate (PET) fiber were manufactured using the eco-friendly materials, $BaSO_4$ and $Bi_2O_3$. Yarns were fabricated by the melt spinning process, and fabrics were woven. With 5 wt% of shielding material, the yarns' shield against radiation and was sufficiently strong for fabric weaving. The fibers' radiation shielding averaged 9–13%, with the $Bi_2O_3$ fiber displaying better shielding performance than the $BaSO_4$. It is believed that the findings of this study on improving the yarn manufacturing process could be applied for protection against low-dose and scattered rays in medical applications and for aerospace radiation protection. In addition, the proposed shielding fibers' flexibility makes them suitable for future use in the production of various radiation shields.

**Keywords:** medical radiation; radiation exposure; general protective clothing; apron; hospital protection

## 1. Introduction

Medical culture, which has traditionally been treatment-oriented, is changing to a culture focused on early diagnosis and prevention. Accordingly, the use of medical radiation is gradually increasing in both range and frequency [1,2]. In medical institutions, radiation shielding is generally applied passively to the areas not being exposed to radiation for diagnosis and treatment. Effective shielding is difficult because of the wide range of doses, which differ according to the area being inspected [3,4]. Recently, interest in active shielding has been increasing in the medical field as well as in industry as the use of radiation has increased. In active shielding, the radiation shield is subdivided and utilized according to the exposure dose and the intensity of the radiation energy [5]. Radiation shields include shielding walls, shielding curtains, shielding garments, thyroid protection equipment, and gloves, as well as radiation-shielding products that are manufactured and utilized for various purposes [6,7]. Existing radiation protection products generally accomplish shielding by using Pb. However, due to the risks associated with Pb, research toward the development of eco-friendly materials has been increasing in recent years [8–10]. For reasons of economy and workability, most radiation shields still use Pb as their main material. However, in the case of Pb aprons, fine cracks may occur depending on the method of use or storage. Therefore, it is becoming increasingly important to address the challenge of providing shielding clothing with flexibility.

The intensity of radiation energy that is generated in medical applications is mostly within a certain range; radiation shielding is utilized by categorizing the direct and indirect radiation regions according to the energy intensity [11]. Most current radiation-shielding equipment provide protection that is equivalent to a Pb thickness of 0.25 to 0.5 mm [12].

Radiation shields that are used in medical institutions are typically used only briefly. Nevertheless, the lightness of the shield is an important consideration for increasing user activities [13,14]. Radiation shielding is usually applied for protection against ionizing radiation. For this purpose, the higher the atomic number of the material constituting the shield and the thicker the shield, the greater the shielding efficiency [15]. In medical institutions, there is an increasing interest in using non-Pb shielding sheets and shielding fibers for protection against radiation. In reality, however, radiation-shielding sheets and fibers have insufficient shielding performance to replace Pb [16].

Radiation shields are traditionally used in the form of films, sheets, or fibers manufactured using Pb powder or mixing it with a polymer compound [17]. In order to minimize the weight of the shields, various parameters, such as the conditions for mixing with the polymer and the density, mass, and number of molecules of the shielding material, must be considered. Technical conditions, such as those of the manufacturing technology, must also be considered [18].

The weight of radiation-shielding aprons used in medical institutions ranges from approximately 2.9 kg to 3.5 kg; this heaviness limits the activities of medical staff and patients [19]. The most suitable material for developing lightweight shielding clothing is radiation-shielding fiber. However, shielding fibers fail to exhibit efficient shielding performance because of limitations in the manufacturing process and pinholes in the fabric [20]. Therefore, when the shielding performance of the shielding fiber is interpreted as the intensity of the effective radiation energy, it has a shielding characteristic that can be used in the scattered ray generating region, which is an indirect ray region, rather than the direct ray with high energy intensity [21].

Thus, in future, the radiation energy intensity may be used as a very useful component when manufacturing a laminated composite sheet using shielding fibers.

The existing shielding fiber manufacturing process involves the coating a polymer resin compound, blended with a shielding material, on a nonwoven fabric. This method is difficult to commercialize because of the lower uniformity of coating thickness and the reduced reproducibility of shielding performance [22,23]. In addition, because the proposed method involves combining fiber fabrics, such as nonwoven fabrics, on the shielding sheet, the sheet has a higher rigidity than flexibility, which is an inherent characteristic of fibers.

Therefore, in this study, a flexible shielding fiber fabric was woven and its characteristics were evaluated. This study investigated the use of the melt spinning process during yarn processing, instead of the existing wet spinning process, with the aim of suggesting a method that would facilitate mass production and simplify the process technology. Barium sulfate and bismuth oxide were selected as shielding materials. Two fabrics were woven, each with a yarn containing one of the shielding materials, and the characteristics of the yarns and the shielding performance of the fabrics were compared. The study aimed to ascertain whether a shielding fabric, containing a radiation-absorbing material, can be woven using a process simpler than the previously proposed lyocell process [24]. This paper presents the fabrication method and characteristics of the shielding fiber that enabled user activity with convenience, and it is expected that the fiber has the potential to be adopted for shielding against low-dose radiation.

Shielding fibers using radiation-impermeable substances, as proposed in this study, can be incorporated into clothing for shielding against both scattered and low-dose radiation generated in medical applications and into garments for shielding against natural radiation in the aerospace industry. In addition, they can be a useful component of laminated composite sheets, manufactured using shielding fibers.

## 2. Materials and Methods

Existing shielding fibers are manufactured using a method of coating a polymeric resin compound mixed with a shielding material onto a nonwoven fabric; the flexibility of the shielding fibers depends on the thickness of the coating. There is also a problem with fine cracks appearing on the surface and inside the shielding sheet because of wear on the coated parts. To address this problem, in this study, polymer masterbatches of fibers (Figure 1) were developed by mixing barium sulfate and bismuth nanoparticles of size 100 nm (Sigma-Aldrich) with polyethylene terephthalate (PET). In addition, to reduce the moisture in the developed masterbatch to less than 20 ppm, a dryer zone was set, and the masterbatch was mixed with PET chips to perform vacuum drying for 36 h.

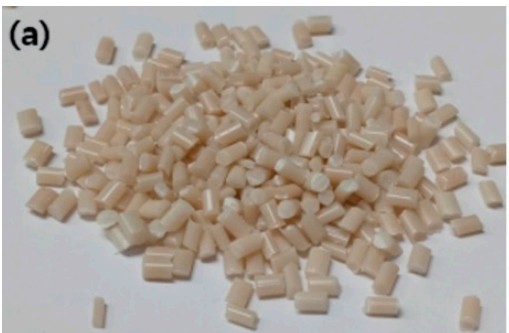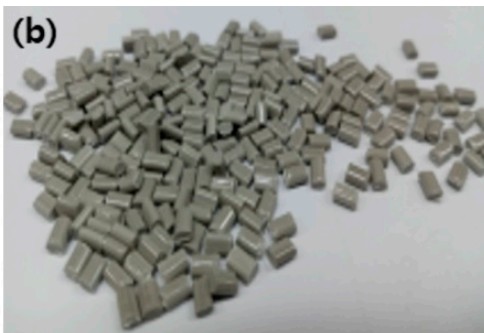

**Figure 1.** Polymer masterbatches for shielding fiber: (**a**) barium sulfate (50 wt%) masterbatch and (**b**) bismuth oxide (20 wt%) masterbatch.

The yarn, used for the shielding fiber fabric, was fabricated using the existing lyocell process. Normally, it is manufactured in a liquid state to which a spinning solution containing cellulose is added. However, this process has problems, such as the need for removal of the solution in the final process and adjustment of the strength of the yarn in the drawing process, which makes mass production difficult [25,26]. Therefore, in this study, the yarn for the radiation-shielding fibers was fabricated under the conditions listed in Table 1, by using a melt spinning process in order to reduce the problems encountered in wet spinning process.

**Table 1.** Conditions for melt spinning yarn for shielding fiber.

| Condition | Adjustable Value |
| --- | --- |
| Content of fine particles in fiber (wt%) | 5 |
| Extruder temperature range (°C) | 300–315 |
| Extruder pressure (Pa) | $1.5 \times 10^{-4}$ |
| Spinning pack temperature (°C) | 292 |
| Spinning pack pressure (Pa) | $1.5 \times 10^{-3}$ |
| Cooling time (s) | 0.43 |
| Take-off roll speed (rpm) | 3230 |

A masterbatch, containing a shielding material, was mixed with PET at a temperature of 290 °C. In this way, a yarn, containing 5 wt% barium sulfate or bismuth oxide, was produced by the melt spinning process. The content of barium sulfate and bismuth oxide in each yarn was analyzed by energy-dispersive X-ray spectroscopy (EDS) (Hitachi, HD-2300, High-Tech Corp, Tokyo, Japan). In addition, the dispersal of the shielding material was observed with an optical microscope (field emission scanning electron microscope (FESEM), Hitachi, S-4800, High-Tech Corp, Tokyo, Japan) after thin-film sectioning with a microtome (Leica, RM2235).

The process of applying the melt spinning process in a solid state consisted of four steps (Figure 2), by which two types of yarn for shielding fibers, containing barium sulfate

or bismuth oxide, respectively, were produced. In order to investigate the characteristics of the fabricated yarns, the fineness, tensile strength, and elongation at break were evaluated. In this process, the yarn must have sufficient strength so that there are no breakage issues, even when the material is used in weaving textile fabrics.

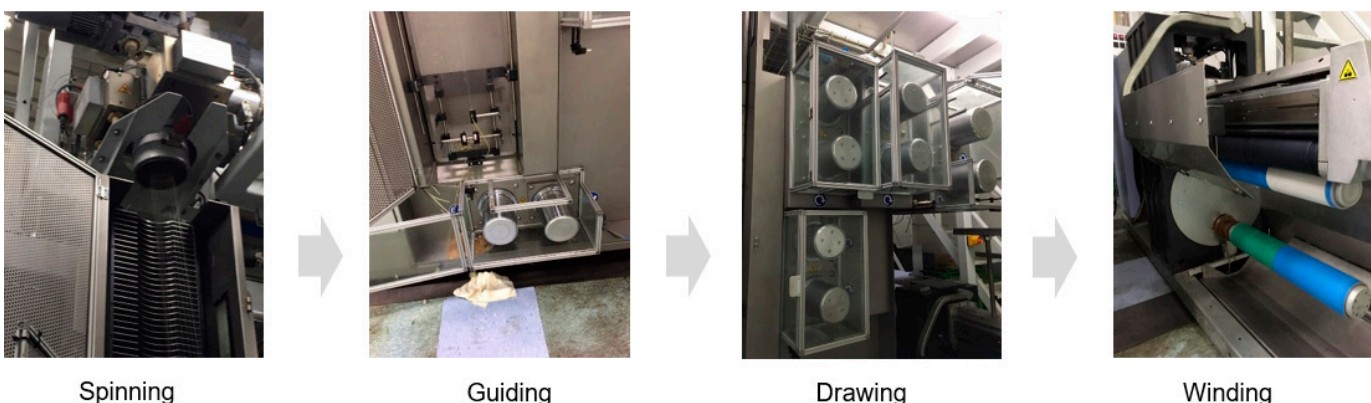

Spinning      Guiding      Drawing      Winding

**Figure 2.** Process for fabrication of yarn for shielding fiber.

The fabric using the shielding fiber yarn was woven at a high density using the method shown in Figure 3. Usually, the shielded fibers are laminated or the shielding materials are dispersed twice (double dispersal) to improve shielding performance. In this study, neither of these methods was applied. Therefore, the shielding performance and characteristics were compared and evaluated for only one sheet of each fabric. In addition, in the weaving process, the mineral content was adjusted to increase the weave density. The optimal condition for the textile fabrication method is the plain weave method. However, in this study, the twill weave method was applied through several experiments. This method is based on the plain weave method, but since the ratio between the weft and warp threads is changed, weaving at high density is possible. When the fiber fabric is compressed to form a laminated structure, improved shielding performance can be expected, but flexibility of the shielding fiber cannot be expected.

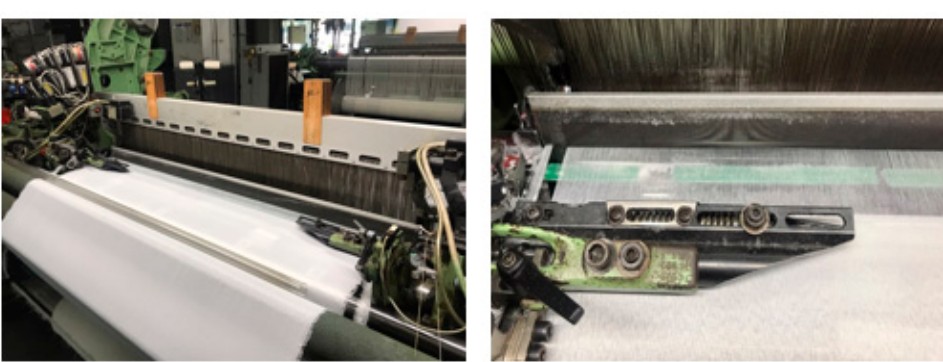

**Figure 3.** Process for fabrication of shielding fiber (rapier loom, Dornier, HTVS 8/S20).

The intensity of radiation energy attenuates as it passes through the shielding fiber. As a result, the shielding fiber absorbs energy, and therefore, the X-ray linear attenuation coefficient is required to calculate the absorbed energy. The linear attenuation coefficient is a value indicating the extent to which the radiation is attenuated; it is in units of $cm^{-1}$. The equation for obtaining the linear attenuation coefficient, $\mu$, is as follows:

$$\mu = \frac{In\left(\frac{I_0}{I}\right)}{t} \tag{1}$$

where $I_0$ is the dose incident on the shield, $I$ is the dose transmitted through the shield, and $t$ is the shield thickness (cm).

The shielding performance of radiation shields, used in medical applications, is often indicated in terms of lead equivalency (mmPb). The lead equivalent is the thickness of Pb that would provide the same shielding effect as the shield when it is irradiated with radiation. The shielding performance of a shielding fiber differs significantly from that of Pb; therefore, it is difficult to express it in lead equivalence. Consequently, in this study, the same thickness is suggested based on the standard aluminum equivalence [27]. Al is mainly used for filtering high energy X-rays in X-ray generators. This study, after measuring the shielding performance while changing the thickness of the Al filter from 0.1 mm to 4 mm, proposed a formula for calculating the aluminum equivalent, $D_{Al}$ (mmAl):

$$D_{Al} = \frac{\mu_F}{\mu_{Al}} \times D_F, \tag{2}$$

where $\mu_{Al}$ represents the linear attenuation coefficient of Al, $\mu_F$ represents the linear attenuation coefficient of the shielding fiber, and $D_F$ represents the thickness of the shielding fiber.

Thus, in this experiment, the shielding performance of the fabricated shielding fiber was evaluated based on Equations (1) and (2).

In order to change the radiation used in this experiment into effective energy, which is a single energy, $\log_e$ is taken on both sides in the attenuation exponential law($I = I_0 e^{-\mu\chi}$, where $\chi$ is the attenuation thickness), and the slope is calculated from the graph of $\mu = -\alpha\chi$ to measured the half-value layer. In addition, after obtaining the value of the linear absorption coefficient $\mu$ from this slope, it could be calculated as the half-value layer = $0.693/\mu$. Finally, effective energy was calculated using Hubbell's mass absorption coefficient table [28] in order to calculate the effective energy having the same half-value layer as the single energy corresponding to the single energy from the previously obtained half-value layer.

The experimental method for testing the shielding performance of the two types of shielding fiber was the geometric condition shown in Figure 4, and the shielding rate of the shielding fibers was calculated as $(1 - W/W_0) \times 100$ [29], where $W$ is the dose measured when there is shielding fiber between the X-ray tube and the dosimeter, and $W_0$ is the dose measured when there is no shielding fiber between the X-ray tube and the dosimeter. All values were tested 10 times using an X-ray generator (Toshiba E7239, 150 kV–500 mA, Tokyo, Japan), and the average value was used. The dose detector was a DOSIMAX *plus* I (IBA Dosimetry), which was used after inspection and calibration.

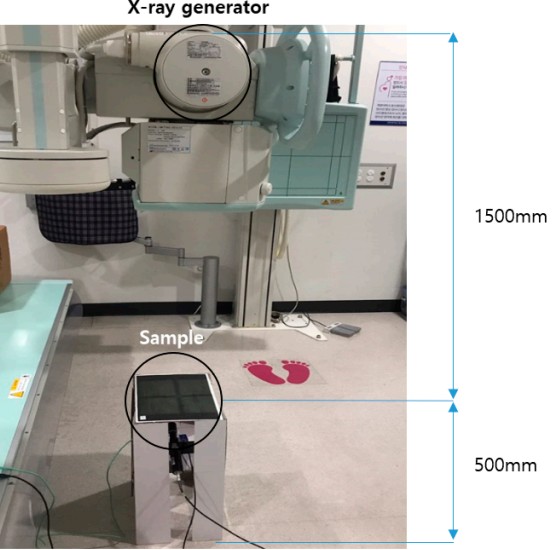

**Figure 4.** Experimental setup for evaluating the performance of the shielding fiber.

## 3. Results

### 3.1. Radiation-Shielding Yarn

The properties of the yarns produced using bismuth oxide and barium sulfate, which are radiation-shielding materials, are shown in Figure 5. It can be seen that the particles of shielding material contained in the yarns are very small and are distributed within the yarns. Therefore, the fibers can be woven while properly maintaining the flexibility and strength of the yarn. In addition, in order to show the shielding performance, the best result was obtained with 5 wt% of the shielding material in the yarn.

| Yarn Containing BaSO$_4$ | Yarn Containing Bi$_2$O$_3$ |
|---|---|

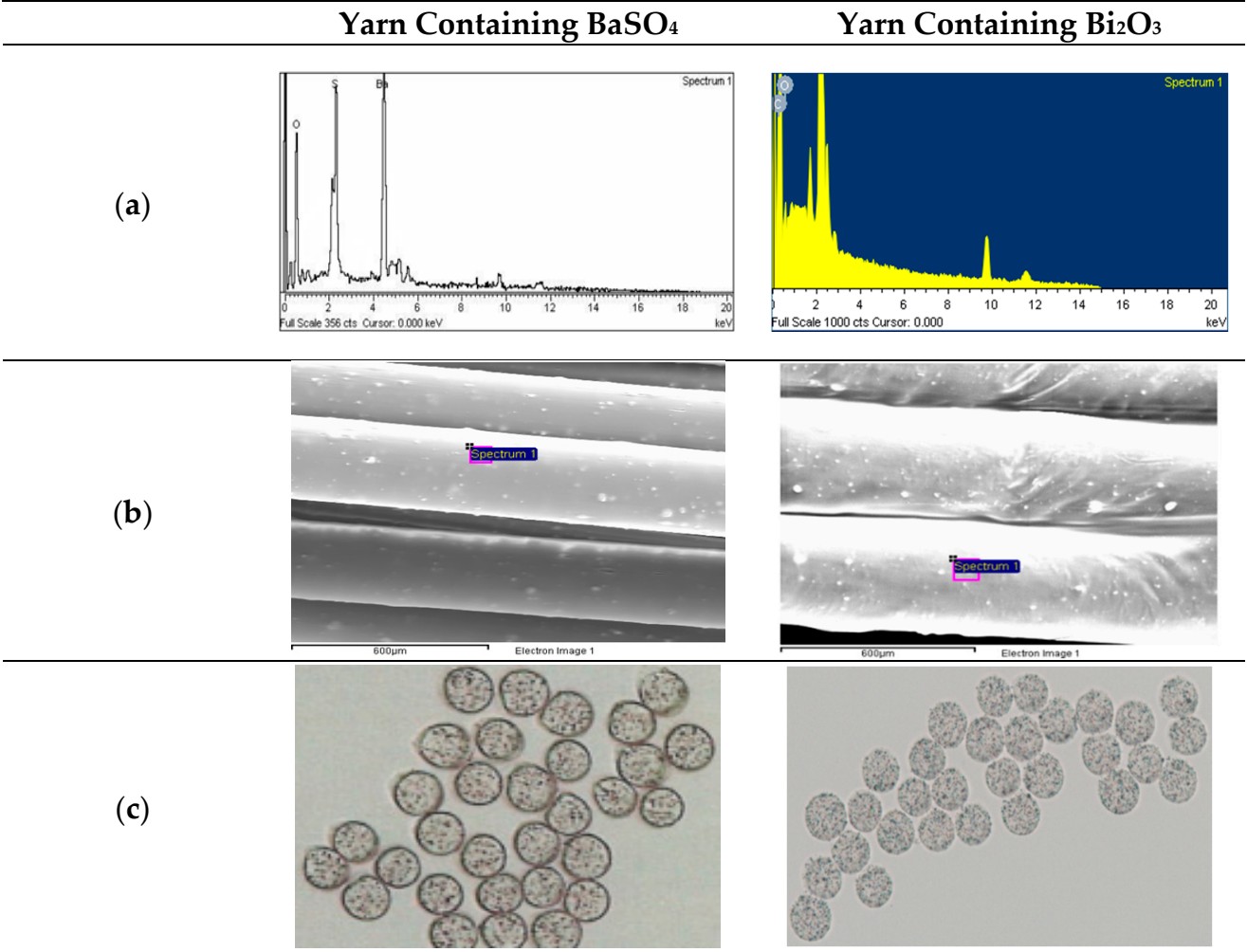

**Figure 5.** Radiation-shielding fiber yarn analysis: (**a**) results of energy-dispersive X-ray spectroscopy (EDS) analysis of yarn components, (**b**) cross-section SEM images (400×), and (**c**) surface SEM images (600×).

The characteristics of the yarns woven using this process are listed in Table 2. The tensile strength of the yarns containing barium sulfate and bismuth oxide was 3.14 g/d and 4.11 g/d, respectively, which are suitable values for weaving radiation-shielding fibers; the yarn with the bismuth oxide had the higher value. The measured value of elongation at break was higher than that of a typical textile intended for use as a general clothing material; this is because of its high mineral content. In addition, in the twisting process preparatory to the weaving process, weaving was performed after processing under low twist-per-meter (TPM) conditions, and the problem of yarn breakage could be prevented.

**Table 2.** Physical properties of shielding fiber yarns.

| Property | Yarn Containing $BaSO_4$ | Yarn Containing $Bi_2O_3$ |
|---|---|---|
| Fineness (denier) | 75.1 | 76.2 |
| Tensile strength (g/d) | 3.14 | 4.11 |
| Elongation at break (%) | 29.4 | 69.5 |
| Mineral content (wt%) | 5 | |
| Twists per meter (TPM) | 75 denier/36 filaments | |

Denier is the thickness unit of the fiber; it is 1 d is when a 9000 m thread weighs 1 g. Filament is the number of yarn in the minimum unit. TPM (twist-per-meter) is the number of twists given to a 1 m thread.

### 3.2. Radiation-Shielding Fabric

The fabrics were woven under the conditions shown in Figure 6, and the characteristics of the fabrics were analyzed, with the results as given in Table 3. The tensile strength values for the two fabrics were similar, but the elongation at break and tearing strength values were higher for the bismuth oxide fabric. This result was because the content of the bismuth oxide in the masterbatch produced before the melt spinning process was lower than that of barium sulfate. That is, the PET content affects the physical properties of the shielding fiber fabric.

| | Yarn | Fabric | SEM: Surface (50×) | Thread Count (Threads/Inch) | |
|---|---|---|---|---|---|
| | | | | Warp | Weft |
| Containing $BaSO_4$ | | | | 200 | 100 |
| Containing $Bi_2O_3$ | | | | 150 | 74 |

**Figure 6.** Conditions for weaving of the fabrics.

**Table 3.** Characteristics of the shielding fiber fabrics.

| Fabric Type | Tensile Strength (N) | | Elongation at Break (%) | | Tearing Strength (N) | | Specific Weight (g/m$^2$) | Thickness (mm) |
|---|---|---|---|---|---|---|---|---|
| Containing $BaSO_4$ | Warp | 740 | Warp | 34 | Warp | 16 | 114–118 | 0.19–0.20 |
| | Weft | 290 | Weft | 20 | Weft | 9 | | |
| Containing $Bi_2O_3$ | Warp | 710 | Warp | 58 | Warp | 28 | 110–120 | 0.20–0.21 |
| | Weft | 590 | Weft | 76 | Weft | 18 | | |

### 3.3. Radiation-Shielding Performance

The results for the radiation-shielding performance of one layer of each of the two fiber fabrics are shown in Table 4. In terms of the effective energy of radiation, the shielding rate was high in the low-energy region. However, in the monoenergy region, where the tube voltage was high, no observable difference was seen and the shielding rate was less than 10%. Therefore, it can be inferred that an effective response can be obtained in the low-dose scattered-ray shielding area. Overall, the shielding performance of the fabric containing bismuth oxide was superior to that of the fabric containing barium sulfate. From the measurements of the Al equivalent thickness ratio using the Al filter, fabrics containing barium sulfate provided shielding equivalent to an Al thickness of 0.18–1.48 mm, and

fabrics containing bismuth oxide showed the equivalent of 1.52–3.24 mm. These results are consistent with the graph of radiation transmittance by Al thickness, shown in Figure 7. Thus, it was determined that the fabric containing the bismuth oxide provides higher shielding performance. The radiation-shielding rates of the two shielded fibers developed in this study represent an average of 9–13%, a range appropriate for application in shielding such as for low-dose radiation or scattered rays.

**Table 4.** Shielding performance of the shielding fiber fabrics.

| Radiation Type | Effective X-Ray Energy (keV) | Radiation-Shielding Rate (%) | |
| --- | --- | --- | --- |
| | | Containing $BaSO_4$ | Containing $Bi_2O_3$ |
| X-ray | 25.2 | 13.8 | 21.5 |
| | 27.9 | 11.5 | 18.4 |
| | 31.4 | 9.4 | 12.8 |
| | 42.8 | 8.6 | 10.2 |
| | 50.8 | 7.4 | 9.4 |
| | 57.2 | 4.1 | 7.1 |

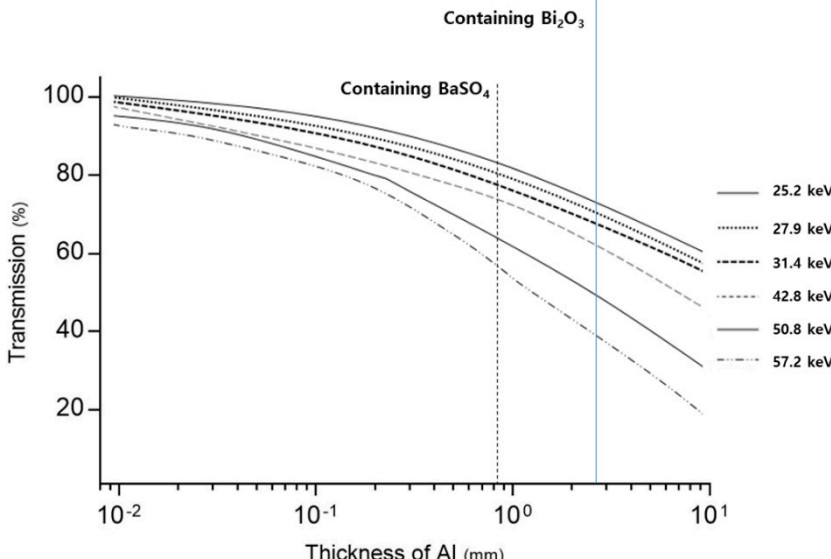

**Figure 7.** Radiation transmittance according to Al thickness.

## 4. Discussion

The most important consideration of radiation-shielding clothing made using shielding materials with X-ray absorption characteristics is the activities of the wearer. Radiation-shielding clothing currently used in medical applications and in industry imposes many restrictions on user activities because of its heavy weight. Therefore, various studies have been conducted to discover new shielding materials and develop manufacturing process technologies for the production of lightweight shields [30–32].

The most meaningful aspect of this study is that it proposes a specific manufacturing process for mass production. The yarn containing the existing shielding material was manufactured from organic and inorganic composite fibers, and their manufacture employed the wet spinning process by using a cellulose solution through a lyocell process. Although that method may increase the mineral content, it is difficult to use for mass production due to the solvent removal stage in the fabrication process [33–36]. The conditions for spinning yarn that maintains a constant strength during the process of weaving shielding fiber are difficult. If the shielding material content—that is, the amount of mineral—is increased, the shielding performance will be excellent, but the tensile strength will be weakened, which can cause problems in weaving. Conversely, if the mineral content is reduced to facilitate

weaving, the shielding performance will be reduced. Therefore, the optimum conditions must be determined.

In this study, yarns of PET fiber were fabricated using barium sulfate and bismuth oxide and with a multifilament of 75 denier/36 filaments using a melt spinning process. The experimental results show that it was only possible to fabricate yarn up to a shielding material content of 5 wt%, and increasing the shielding material content reduced the tensile strength of the yarn. Therefore, further investigation of the melt spinning process and masterbatch manufacturing conditions are needed in future studies. Using the master-batch for yarn production is one way to increase the content of the shielding material, and the multifilament structure during melt spinning process is used for the same purpose. The shielding performance of shielding fibers is ultimately determined by the content of shielding material and the process technology. The reproducibility of the shielding performance of the shielding fiber and maintenance of physical properties are very important considerations. Therefore, rather than simply investigating ways to increase the shielding material content, there is a need to research methods that can improve shielding performance through process technologies, such as weaving technology.

Existing shielding fibers are manufactured by coating a fabric with a polymer composite—elastomer, rubber, and a mixture of tungsten, barium sulfate, and bismuth oxide. This method has the disadvantage of not imparting the inherent properties of the fibers to the fabric because of their low flexibility and high rigidity [37–39]. In addition, it is difficult to reproduce the shielding performance because the shielding performance must be controlled by adjusting the coating thickness during the process [40,41]. However, when a shielding material is added to the yarn itself, this problem can be solved, enabling the same shielding performance to be obtained. In addition, to improve the shielding performance, it may be advantageous to consider a method in which a shielding fiber fabric is laminated or a shielding material is dispersed on the fabric twice.

The shielding fiber developed in this study is intended for use as an additional component of composites such as for scattered-ray-shielding clothing or shielding sheets. As the proposed radiation-shielding fiber is light and flexible, it can be used for the purpose of shielding the thyroid gland and the gonads through the manufacture of uniforms for aviation workers, and it can also be used as a composite fabric material for fire suits [42–44]. It is particularly suitable as a material to protect aviation crew members against natural radiation, and it can also be used as a packaging material for precision machinery. In addition, it is possible to improve the shielding performance by stacking layers or adding a neutron-shielding material. This can be proposed as a low-dose defense shielding material that does not limit the activity of the wearer. In particular, for use in medical institutions, a shielding unit tailored to the intensity of radiation energy can be manufactured for the medical personnel working at a given distance from the radiation-generating area [45].

Radiation-shielding suits are unfavorable in terms of cost when manufactured with eco-friendly materials except lead, which poses a risk. Also, in the case of textile weaving as in this study, flexibility and activity are guaranteed, but the economic aspect is still disadvantageous. However, it has an important meaning because it can improve the activities of medical personnel and aviation crew members and allow escape from dangers to health. Although this study did not find substitute materials for Pb because various shielding materials were not used, it introduced the possibility of the mass production of shielded fibers containing bismuth oxide and barium sulfate, which are the most commonly used Pb substitutes in shielding materials.

## 5. Conclusions

In order to fabricate a radiation-shielding fiber, yarns were prepared using barium sulfate and bismuth oxide as shielding materials. The two shielding fibers produced in the study displayed an average shielding performance of 9–13% in terms of the reduction in X-ray energy intensity, with the fabric containing bismuth oxide showing the better results. The content of each shielding material applied to the yarn was 5 wt%, and the

thickness of the two shielding fibers was 0.2 mm. The shielding fiber proposed in this study is suitable for use in shielding against scattered rays in medical institutions and against natural radiation in the aerospace industry.

**Funding:** The APC was funded by the National Research Foundation of Korea, grant number 2020R1I1A3070451.

**Institutional Review Board Statement:** Not applicable.

**Informed Consent Statement:** Not applicable.

**Data Availability Statement:** Data is contained within the article.

**Conflicts of Interest:** The author declares no conflict of interest.

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
