# Peer review of "Construction of a Medical Radiation-Shielding Environment by Analyzing the Weaving Characteristics and Shielding Performance of Shielding Fibers Using X-ray-Impermeable Materials"

_applsci, doi:10.3390/app11041705_

Round 1

Reviewer 2 Report

Regarding radiation shielding, the authors examined the construction of a medical radiation shielding environment by analyzing the weaving characteristics and shielding performance of shielding fibers using X-ray impermeable materials. The text is well written, and the findings and experimental approaches presented seem to be solid.

Please correct "Fig-ure" of L101.

Now, when you think that you will actually make a radiation shielding apron etc. with this newly developed material, in reality, you have to consider the cost aspect as well. I would like to ask you to describe from this point of view in the discussion.

Round 2

Reviewer 1 Report

The authors has done an excellent and thorough job of revising the paper, and I appreciate the author's diligence.

May I suggest the following sentence (or something along these lines) in Section 3.3, at the end of the paragraph that ends on line 242:   “The averages are calculated using the radiation shielding rates for the six effective energy values listed in Table 4 corresponding to the barium oxide and bismuth oxide based fabrics respectively."  This will make your conclusions abundantly clear.

I agree that the paper should be published, even if the author decides not to add a sentences like I suggest.